# Diversity of Culturable Alkaliphilic Nitrogen-Fixing Bacteria from a Soda Lake in the East African Rift Valley

**DOI:** 10.3390/microorganisms10091760

**Published:** 2022-08-31

**Authors:** Yordanos Ali, Addis Simachew, Amare Gessesse

**Affiliations:** 1Institute of Biotechnology, Addis Ababa University, Addis Ababa P.O. Box 1176, Ethiopia; 2Industrial Biotechnology Research Directorate, Bio and Emerging Technology Institute, Addis Ababa P.O. Box 5954, Ethiopia; 3Department of Biological Sciences and Biotechnology, Botswana International University of Science and Technology, Private Bag 16, Palapye 10071, Botswana

**Keywords:** nitrogen fixation, alkaliphile, Chitu, soda lake, *Spirulina*, *Arthrospira*

## Abstract

Lake Chitu is a highly productive soda lake found in the East African Rift Valley, where *Arthrospira fusiformis* (*Spirulina platensis*) is the main primary producer. High biomass accumulation requires an adequate supply of nitrogen. However, Lake Chitu is a closed system without any external nutrient input. A recent study has also demonstrated the presence of a diverse group of denitrifying bacteria, indicating a possible loss of nitrate released from the oxidation of organic matter. The aim of this study was to isolate culturable nitrogen-fixing alkaliphiles and evaluate their potential contribution in the nitrogen economy of the soda lake. A total of 118 alkaliphiles belonging to nine different operational taxonomic units (OTUs) were isolated using a nitrogen-free medium. Nineteen isolates were tested for the presence of the *nifH* gene, and 11 were positive. The ability to fix nitrogen was tested by co-culturing with a non-nitrogen-fixing alkaliphile, *Alkalibacterium* sp. 3.5*R1. When inoculated alone, *Alkalibacterium* sp. 3.5*R1 failed to grow on a nitrogen-free medium, but grew very well when co-cultured with the nitrogen-fixing alkaliphile NF10m6 isolated in this study, indicating the availability of nitrogen. These results show that nitrogen fixation by alkaliphiles may have an important contribution as a source of nitrogen in soda lakes.

## 1. Introduction

Soda lakes of the East African Rift Valley are unique habitats characterized by stable, high pH in the alkaline range, and high salinity. Alkalinity is a result of high carbonate-bicarbonate concentration, which also provides an unlimited carbon dioxide reserve for photosynthesis. In addition, because of their tropical location, these lakes get unlimited light all year round, and experience little temperature fluctuation. As a result, these lakes are highly productive, with the highest primary production reported for any natural habitat [1,2]. High primary production also implies an abundance of organic matter released due to cell death or excess photosynthesis, which, in turn, supports a diverse group of heterotrophic microorganisms [3].

Lake Chitu is one of the soda lakes found in the central Rift Valley of Ethiopia. It is a crater lake with a stable pH of around 10.5 and salinity of about 6% [4,5]. The blue-green algae, *Arthrospira fusiformis* (formerly known as *Spirulina platensis* or commonly as Spirulina), is the most dominant primary producer in the lake [4]. Oxygen measurement showed that, depending on the biomass concentration on the surface and the mixing rate, below a depth of 1–3 m (and sometimes below a depth of half a meter) from the surface, the lake is totally anoxic. Therefore, throughout the year, a significant portion of the lake’s depth is anoxic. 

A study using culture-independent molecular techniques revealed the presence of high microbial diversity in Lake Chitu. Furthermore, both microbial diversity and abundance increased with increasing depth [5], which is probably a result of an increased availability of organic matter released from dead algal cells sinking downward. This could also indicate that the available organic matter is utilized by a diverse group of microorganisms distributed along the entire depth of the lake. Such microbial degradation of organic matter involves hydrolysis of macromolecules, such as proteins, DNA, RNA, lipids, and carbohydrates to their respective monomeric units, such as amino acids, fatty acids, and simple sugars. Further oxidation of these monomeric units under aerobic and anaerobic conditions could lead to their conversion to CO_2_, nitrate, and sulfate.

The main primary producer in Lake Chitu, *Arthrospira fusiformis,* is well known for its high protein content [6]. During its active growth cycle, the entire lake appears covered by a thick green biomass, indicating active growth. Dense algal growth requires an unlimited supply of all the required nutrients and an availability of light energy. Since Lake Chitu has a high carbonate bicarbonate concentration and is found in the tropics, the supply of CO_2_ and access to energy (sun light) could not be a limiting factor on the rate of photosynthesis. However, in addition to the above inputs, an accumulation of a large quantity of biomass also requires an adequate supply of nitrogen sources for the synthesis of amino acids, nucleotides, and other nitrogen-containing organic compounds. 

The lake is essentially a closed basin with no surface inflow and outflow. The only known inflows are a few hot springs that spring from the shore of the lake, and drain directly into the lake. Therefore, the lake does not get any external nitrogen supply, be it in the form of organic or inorganic nitrogen. One assumption that could be put forward is a closed nitrogen cycle within the lake, where ammonia is released upon the degradation of organic matter (for example, proteins, DNA, and RNA), and is released as ammonia and nitrate, which will then be taken up by the primary producer and other heterotrophic microorganisms in the lake. 

However, the assumption of a closed nitrogen cycle within the lake could only be true in the absence of denitrification. A recent study in our laboratory on Lake Chitu demonstrated the presence of a diverse group of denitrifying bacteria, many of them exhibiting a high efficiency of denitrification [7]. Since the bulk of the water column (up to 95%) is devoid of oxygen, the presence of a diverse group of denitrifying prokaryotes may indicate that the nitrate and nitrite released during the degradation process of nitrogen-containing organic compounds could serve as a final electron acceptor, leading to its release as molecular nitrogen, N_2_. If nitrate and nitrite of the lake are lost as N_2_ through the process of denitrification, the question regarding the source of nitrogen to support prolific cyanobacterial growth in such a closed environment remains unanswered.

Therefore, if nitrate is lost due to denitrification, the other potential source of nitrogen to support algal growth in this closed environment could be nitrogen fixation. Biological nitrogen fixation involves the reduction of N_2_ into ammonia (NH_3_), the only inorganic nitrogen that can be assimilated by living organisms. Only a few groups of prokaryotes belonging to the domains, archaea and bacteria, can fix atmospheric nitrogen. These diazotrophic microbes possess the enzyme, nitrogenase, a complex biocatalyst consisting of several subunits that are involved in nitrogen fixation. Nitrogen-fixing microorganisms could exist as free-living organisms in terrestrial and aquatic habitats, or form symbiotic associations with other organisms with varying degrees of complexity, and are a vital source of nitrogen in any ecosystem [8]. Therefore, the nitrogen economy of Lake Chitu might depend on biological nitrogen fixation.

Most known nitrogen-fixing prokaryotes, whether free-living or symbiotic, grow under neutral pH and in the presence of low or moderate concentrations of salt. On the other hand, the environment in Lake Chitu is alkaline and saline. Therefore, prokaryotes that exist in this lake are subjected to double extreme conditions. The aim of this study was, therefore, to investigate the presence and diversity of culturable nitrogen-fixing alkaliphiles and their potential role in the nitrogen economy of the soda lake habitat.

## 2. Materials and Methods

### 2.1. Sampling Site and Sample Collection

Lake Chitu is located in the central Rift Valley of Ethiopia at 7°24′0″ N and 38°25′0.02″ E. It is a small crater lake with a surface area of approximately 800 m^2^, a maximum depth of 17 m, and salinity of about 6%. A few hot springs emerge from the shore and drain into the lake. Other than these hot springs and direct precipitation, the lake is a closed basin with no outflow and inflow.

Water and sediment samples were collected in triplicate from different depths, including the surface (0 m) and the sediment from a depth of 17 m. At the time of sampling, the transition between aerobic and anaerobic zones was determined to be at 2.9 m, and this was selected as one of the sampling depths. The other sampling depths include surface sample (0 m), which represents the oxic zone, 5, 10, 14, and 17 m, all representing the anoxic zone. 

Water and sediment samples were collected using a bottle sampler and an Ekman grab, respectively. Samples were immediately transferred to sterile bottles and tubes, and transported to the laboratory in an icebox kept at 4 °C. Physicochemical parameters, including pH, salinity, conductivity, and temperature, were measured on site at the time of sampling.

### 2.2. Enrichment Cultures

To isolate nitrogen-fixing bacteria, a nitrogen-free media was prepared following the method of Jensen [9] with slight modification. The medium was composed of (g/L): glucose, 5; NaCl, 5; MgSO_4_·7H_2_O, 0.2; K_2_HPO_4_, 0.1; CaCl_2_·2H_2_O, 0.26; FeSO_4_, 0.1; Na_2_CO_3_, 10; and trace mineral solution, 10 mL/L. Glucose, Na_2_CO_3_, and trace mineral salt solution were sterilized separately and added to the rest of the medium after cooling to around 50 to 60 °C. The stock trace metal salt solution was composed of (g/L): H_3_BO_3_, 0.03; NaMO_4_·2H_2_O, 0.003; MnCl_2_·4H_2_O, 0.003; ZnSO_4_·7H_2_O, 0.01; CoCl_2_·6H_2_O; 0.01; NiCl·6H_2_O, 0.002; and CaCl_2_·2H_2_O, 0.001. After adjusting the pH to 7.6 using 1M NaOH, the medium was autoclaved at 121 °C for 15 min. For solid media, agar was added to the above media to a final concentration of 20 g/L.

Enrichment was carried out by inoculating 100 mL sterile liquid media with 1 mL of a well-mixed water. For sediment samples, about 1 g was first suspended in 100 mL sterile diluent, and 1 mL of the suspension was used to inoculate 100 mL medium. The culture was incubated at 30 °C for up to seven days under aerobic and anaerobic conditions. 

### 2.3. Isolation of Nitrogen-Fixing Bacterial Strains

The enrichment culture was serially diluted in the range of 10^−1^ to 10^−5^ using filter-sterilized saline solution, pH adjusted to 10. A 100 µL sample was spread onto agar plates containing nitrogen-free medium. Plates were incubated at 30 °C under aerobic and anaerobic conditions until distinct colonies emerged. A total of 180 colonies were randomly picked and purified through repeated streaking. Isolates were kept on nitrogen-rich agar slants at 4 °C for further characterization. 

### 2.4. Co-Culture of Nitrogen-Fixing Bacterial Isolates with Non-Nitrogen-Fixing Isolate

The release of reduced nitrogen in the form of ammonia by the nitrogen-fixing alkaliphiles was measured indirectly through co-culturing with a known non-nitrogen-fixing alkaliphile, *Alkalibacterium* sp. 3.5R*1. When it is alone, *Alkalibacterium* sp. 3.5R*1 was unable to grow in the nitrogen-free medium. However, when it is cocultured with nitrogen-fixing strains, the organism grows well in the nitrogen-free medium, indicating an uptake of reduced nitrogen released in the medium. Another advantage of using *Alkalibacterium* sp. 3.5R*1 was that it has a marked color difference from the cocultured nitrogen-fixing isolates, making it easier to distinguish and enumerate the two when grown mixed on agar plates.

For the coculturing experiment, one of the alkaliphilic nitrogen-fixing isolate, NF10m6, was used. Thus, the nitrogen-fixing isolate, NF10m6, and the non-nitrogen-fixing strain, *Alkalibacterium* sp. 3.5R*1 (JX434738), were inoculated in a liquid nitrogen-free medium both independently (single culture) and the two together (co-culture), and incubated at 30 °C for ten days under anaerobic conditions. A sample of each culture was taken and serially diluted and spread on a nitrogen-rich solid medium and incubated as described above. In each plate, colonies of the nitrogen-fixing and non-nitrogen-fixing strains were enumerated based on their distinct colony color, and expressed as CFU/mL, and the result is expressed as log number of cells/mL.

### 2.5. Genomic DNA Extraction 

Genomic DNA from 118 pure culture isolates was extracted following the modified freeze-thaw DNA extraction method [10]. Few colonies were picked from an agar plate using a sterile loop and re-suspended in 50 μL of TE buffer, pH 8.0 in 1.5 mL Eppendorf tubes. The tubes were in a boiling water bath for 5 min, followed by immediate freezing at −20 °C for 15 min. The boiling and freezing steps were repeated once again. The cell lysate was stored at −20 °C and used for PCR amplification.

### 2.6. PCR Amplification and Amplified Ribosomal DNA Restriction Analysis (ARDRA) 

PCR amplification of the 16S rRNA gene and endonuclease restriction digestion of the amplified 16S rRNA gene were carried out as described previously [11]. For PCR amplification of the 16S rRNA universal bacterial primer sets, B338f (5′-ACTCCTACGGGAGGCAGCAG-3′) and H1542r (5′-TGCGGCTGGATCACCTCCTT-3′) primers [12] were used. Restriction digestion was carried out using the restriction enzyme, TaqI (5′…T^CGA…3′).

### 2.7. Sequencing of Selected Isolates

Twenty representative isolates were selected from each OTU group and refreshed on nitrogen-free media for the extraction of DNA. PCR amplification and sequencing of the 760 bp 16S rRNA gene fragment were carried as described previously [11] using primer sets, A8f and H1542r [12].

### 2.8. PCR Detection and Cloning of nifH Gene

PCR detection of the *nifH* gene was carried on the nineteen representative isolates selected randomly from each OTU group. PCR amplification of the *nifH* gene was carried by a nested PCR approach as described previously [13]. The first round of the nested PCR was carried out using *nifH* primer sets, *nifH4* (*Azotobacter vinelandii* nucleotide positions 546 to 562; 5′-TTY TAY GGN AAR GGN GG-3′) and *nifH3* (*A. vinelandii* nucleotide positions 1018 to 1002; 5′-ATR TTR TTN GCN GCR TA-3′). The second round of the nested PCR was performed using primer sets, nifH1 (*A. vinelandii* nucleotide positions 639 to 655; 5′-TGYGAYCCNAAR GCN GA-3′) and *nifH2* (*A*. *vinelandii* nucleotide positions 1000 to 984; 5′-ANDGCCTCATYTCNCC-3′). The presence and amplification of *nifH* of the gene was detected by running the amplicons on 2.5% agarose gel electrophoresis. Bands, which correspond to the *nifH* gene at around 360 bp, were excised and gel-purified by using a gel Nucleospine purification kit according to the manufacturer’s instructions (Nucleospine extraction kit 2004). The elute was preserved at 4 °C for cloning and further analysis.

### 2.9. Cloning of nifH Gene, Reamplification, and Sequencing of Cloned nifH Gene Products

Depending on the band intensity, 3–4 µL of the gel-purified fragment of the PCR-amplified *nifH* gene (approximately 360 bp) was ligated into the plasmid vector (pCR^®^ 2.1-TOPO^®^ vector, 3.9 kb) using a TOPO^®^ TA Cloning Kit (Invitrogen, Waltham, MA, USA, 2006) according to the manufacturer’s instruction. The heat shock method of transformation on One Shot^®^ Chemically Competent *E. coli* cells (TOPO10) and the screening of transformed cells were carried out as described in TOPO TA Cloning Kit (Invitrogen, 2006).

Cloned PCR products with the right insert were screened by PCR amplification using M13 primers directly from overnight cultured transformed *E. coli* cells, and by running the amplified PCR products on agarose gel (3% *w*/*v*) electrophoresis. Clones with the right insert were sequenced using primer sets, *nifH1* and *nifH2*, from both directions.

### 2.10. Phylogenetic Analysis 

Sequences were edited manually using CLUSTALW [14]. The program, Bellerophon on Detection, of chimeric sequences in multiple alignments was applied to the studied sequences in order to detect possible chimeric artifacts [15]. Reference 16S rRNA sequences were retrieved from the NCBI GenBank database available at http://www.ncbi.nlm.nih.gov (accessed on 1 April 2022) using BLAST (Basic Local Alignment Search Tool) analysis to provide the identity of the sequence. The phylogenetic relationship of sequences to the closest matches in the public database based on 16S RNA gene sequences was constructed by using the maximum likelihood method using distances calculated with the Kimura 2-parameter model [16] in MEGA 10.0 software [17]. The stability and reliability of the relationships of lineages on the inferred trees was tested by bootstrap analysis for 1000 replicates [18]. Sequences with sufficient length were included in the alignment; the sequence alignments were then corrected manually, and sequences with ambiguous alignment positions were removed from the analysis.

### 2.11. Sequence Deposition

The 16S rRNA and *nifH* genes’ nucleotide sequences were submitted to the NCBI nucleotide database under accession numbers, OL444760-OL444777 (16S rRNA gene) and OL606585-OL606586 (*nifH* gene).

## 3. Results

### 3.1. Physicochemical Parameters of the Lake and Enrichment for Nitrogen-Fixing Alkaliphiles

All samples collected from different depths of the lake had a pH over 10, a salinity of about 5.5–5.6%, and conductivity of 63–64 mS/cm^−1^. At the time of sampling, the depth for the oxic–anoxic transition was determined to be at 2.9 m, and the maximum depth was 17 m (Table 1 and Figure 1). The pH, salinity, and conductivity of Lake Chitu reported at different times were, respectively, 10.15, 4.49%, and 49 mS/cm^−1^ in 1994 [4]; 10.2, 4.5%, and 49 mS/cm^−1^ in 1997 [19]; and 10.1, 3.75%, and 56 mS/cm^−1^ in 2014 [20]. In 2013, Lanzen et al. also measured a pH of 10.4 and a salinity of 5.8%. These results show that the pH, salinity, and conductivity of the lake changed little in a nearly two-decade period. 

Enrichment in nitrogen-free media resulted in the growth of a high number of cells from each depth (Figure 1). The highest number of cells in terms of CFU was obtained for the enrichment culture inoculated from the sediment for the sample taken from 10-m depth.

Out of the 180 distinct colonies tested, 118 isolates grew on nitrogen-free medium (Figure 2). Of these, 25 isolates were obtained from sediment sample (NFM) collected from a depth of 17 m. Among water sample isolates, a large numbers of nitrogen-fixing isolates were obtained from samples collected from a depth of 14 m (23 isolates) and 10 m (19 isolates). Thus, more isolates capable of growth on nitrogen-free medium were obtained with increasing depth.

### 3.2. Co-Culture of Nitrogen-Fixing Isolates with the Non-Fixer Isolates

Isolate NF10m6 grew in a nitrogen-free liquid media giving a CFU/mL of 5.2 × 10^3^ after 48-h incubation under anaerobic conditions. On the other hand, *Alkalibacterium* sp. 3.5*R1, an alkaliphile that grows under anaerobic conditions, completely failed to grow in a nitrogen-free medium (Table 2). To check if the nitrogen-fixing isolate, NF10m6, fixes enough nitrogen (N_2_) and makes it available for other organisms, the two strains were co-cultured in the same nitrogen-free medium. As shown in Table 2, both organisms grew when co-cultured, indicating that the nitrogen-fixing bacteria helps the non-nitrogen-fixing *Alkalibacterium* sp. to grow in nitrogen-free broth media by fixing nitrogen by supplying reduced nitrogen for growth.

### 3.3. Diversity of Haloalkaliphilic Nitrogen-Fixing Microbial Community

The ARDRA pattern following restriction digestion using the Taq 1 restriction enzyme clustered the 43 representative isolates into nine operational taxonomic units (OTUs) (Figure 3). Isolates obtained from the sediment samples (17 m) were distributed across five of the nine OTUs, whereas isolates from surface water samples (0 m) were grouped into a single OTU.

### 3.4. Analysis of 16S rRNA Gene Sequence and Construction of Phylogenic Relationship 

Figure 4 shows the phylogenetic relationship of the representative isolates from each ARDRA group based on 16S RNA gene sequences. These isolates were grouped into three clusters, affiliated with the three major bacterial phyla. Thus, 69% of the isolates belong to phylum, *Proteobacteria*; 23% to phylum, *Firmicutes*; and 8% to phylum, *Actinobacteria*. All the isolates affiliated to phylum, *Proteobacteria*, were closely related to alkaliphilic strains of the genus, *Halomonas*, previously isolated from various soda lakes in different parts of the world [21]. Isolates affiliated with phylum, *Firmicutes*, were closely related to alkaliphilic members of genera, *Bacillus* and *Alkalibacterium*, whereas the six isolates affiliated to phylum, *Actinobacteria*, are closely related to genus, *Nesterenkonia.*

### 3.5. Detection and Sequencing of the nifH Gene 

To check if growth in nitrogen-free medium is associated with expression of the nitrogenase enzyme, the possession of the *nifH* gene was checked through PCR amplification. Out of the 19 isolates tested, the *nifH* gene was detected in 11 of the isolates (Figure 5). The *nifH* gene was detected from isolates obtained from all depths, including the surface water.

Out of the 11 bands detected, clone libraries of the *nifH* gene (360 bp) were constructed, out of which, 10 were successfully cloned. Two clones from each clone library (a total of 20 *nifH* gene clones) were picked and sequenced, but only six clones gave good sequences that were used for further analysis (Figure 6). The six *nifH* gene fragments sequenced were found to be closely related to *nifH* gene fragments of uncultured microbial communities of environmental samples (Figure 6).

## 4. Discussion

In this study, alkaliphilic bacterial isolates that grew on a nitrogen-free medium were isolated from samples collected from different depths of Lake Chitu, and were grouped into nine different OTUs. This shows that nitrogen-fixing prokaryotes probably have a significant contribution in the nitrogen economy of the lake. 

Lake Chitu is a crater lake with no inflow and outflow other than a few hot springs that emerge from the shore and enter the lake, and the source of nitrogen to support the high biomass of the primary producer and a diverse group of heterotrophic microorganisms remains unclear. An abundance of denitrifiers, as revealed in a recent study [7], indicates a potential loss of nitrate available in the lake. This is especially important when one considers the fact that a significant portion of the lake (below about 0.5–3 m out of a maximum depth of 17) is devoid of oxygen. Hence, an abundance of denitrifying microorganisms show that nitrate released from the degradation of organic nitrogen could be released as molecular nitrogen. Therefore, to support such prolific growth observed in Lake Chitu, nitrogen fixation by different alkaliphiles might play an important role in supplying the required nitrogen.

In addition to their ability to grow on nitrogen-free medium, the alkaliphilic isolates from Lake Chitu possess the *nifH* gene that code for the nitrogenase enzyme involved in nitrogen fixation. This shows that the alkaliphilic isolates have the genetic potential to fix nitrogen. However, the possession of a nitrogenase gene may not necessarily show the existence of a functional nitrogenase enzyme capable of fixing nitrogen under the alkaline and saline conditions of the lake. In this study, the ability to fix nitrogen was indirectly tested by co-culturing with a known non-nitrogen-fixing bacterial strain which was unable to grow on a nitrogen-free medium. Expression of the nitrogenase gene of nitrogen-fixing prokaryotes is regulated by the availability of nitrogen in the surrounding environment [22]. When the gene is turned on and the cell fixes the atmospheric nitrogen, it is not only fixing enough for itself, but also releases excess nitrogen in the form of ammonia used by other organisms in its vicinity [23], which is also the basis for the symbiotic association between the nitrogen-fixing rhizobia and leguminous plants [24].

The strain used for co-culturing forms a distinct color, different from the nitrogen-fixing isolate, making the enumeration of each organism on the same plate easy. When the non-nitrogen-fixing strain was grown separately on a nitrogen-free medium, no colony was detected. However, upon coculturing with one of the nitrogen-fixing isolates of this study, the strain showed an abundant growth, indicating a supply of fixed nitrogen for the non-nitrogen-fixing strain. This, therefore, indirectly confirms that the alkaliphilic isolate from Lake Chitu expressed the *nifH* gene, and fixed enough atmospheric nitrogen, making reduced nitrogen available for the growth of the cocultured non-nitrogen-fixing organism. This probably indicates that nitrogen fixation by alkaliphilic microbial species may have an important contribution on the nitrogen economy of soda lakes.

The enzyme, nitrogenase, is highly sensitive to oxygen [25]. Thus, nitrogen-fixing prokaryotes either grow anaerobically or develop mechanisms to protect their nitrogenase from exposure to oxygen through different mechanisms. Out of the nine-nitrogen-fixing alkaliphilic OTUs isolated from Lake Chitu, only one OTU was isolated from surface water, whereas eight isolates, accounting nearly 90%, were isolated from the anaerobic zone, below a depth of 2.9 m. Of these, three OTUs were isolated only from the sediment sample at a depth of 17 m. The distribution of the nitrogen-fixing OTUs to the anaerobic part of the lake indicate the adaptation of these organisms to anaerobic life, and this helps to protect the nitrogenase from inactivation by oxygen.

Most of the nitrogen-fixing isolates of the present study are closely related to bacterial strains isolated from various soda lakes in different parts of the world, with 99–100% 16S rRNA gene sequence similarity [21,26,27,28], where members of the genus, *Halomonas*, were the most abundant. Previous studies reported that the genus, *Halomonas*, was the most abundant culturable Gram-negative bacteria in soda lakes and other hypersaline environments [28], able to grow on nutrient-poor culture media. Some members of the genus were also reported to be diazotrophs [29].

The other group of isolates (represented by isolate NF10m6) that grow in nitrogen-free media and possess the *nifH* gene belong to the actinobacterial genus, *Nesterenkonia.* To the best of our knowledge, no member of the phylum, *Actinobacteria*, has been reported to fix nitrogen in soda lake habitats.

The *nifH* gene fragment was amplified from 11 of the 19 isolates tested. Phylogenetic analysis based on *nifH* gene fragment sequences showed that their closest matches were diazotrophic bacterial strains belonging to the phylum, *Proteobacteria*, and uncultured microorganisms isolated from different environments. For example, the *nifH* gene fragment sequenced from clone, Mud6, showed greater than 96% similarity to the *nifH* gene sequence of an uncultured organism clone obtained from the Mediterranean Sea [30]. The *nifH* gene fragments of NF10m6, NF2.9m8, and NF14m10 showed 100%, with a putative nitrogenase reductase gene obtained from the uncultured microorganism clone, B10-31, obtained from a marine habitat [31], whereas the *nifH* gene fragment of isolate, 14m10, showed more than 92% sequence identity with the B21 dinitrogenase reductase (*nifH*) gene of uncultured bacterium clone, DUN1_nifH_+, and uncultured bacterium clone, DUNnif131 (+B26) [32].

The presence of a diverse group of nitrogen-fixing alkaliphiles in soda lakes may indicate the important role these organisms play as sources of nitrogen, supporting the high productivity of soda lakes. Spirulina is the main primary producer in Lake Chitu, a cyanobacterium with a wide range of industrial applications [33]. At present, Spirulina is cultivated commercially using an artificial medium containing an inorganic nitrogen source, mostly in the form of nitrate [34]. In its natural setting at Lake Chitu, Spirulina grows together with nitrogen-fixing prokaryote. This could indicate the interesting possibility of designing a mixed culture system consisting of nitrogen-fixing prokaryotes and the cyanobacteria. In addition to providing a cheap and sustainable way of producing food, animal feed, and other industrial products from the Spirulina biomass [33], such a system could provide an efficient mechanism of fixing CO_2_ released by different industries [34].

## Figures and Tables

**Figure 1 microorganisms-10-01760-f001:**
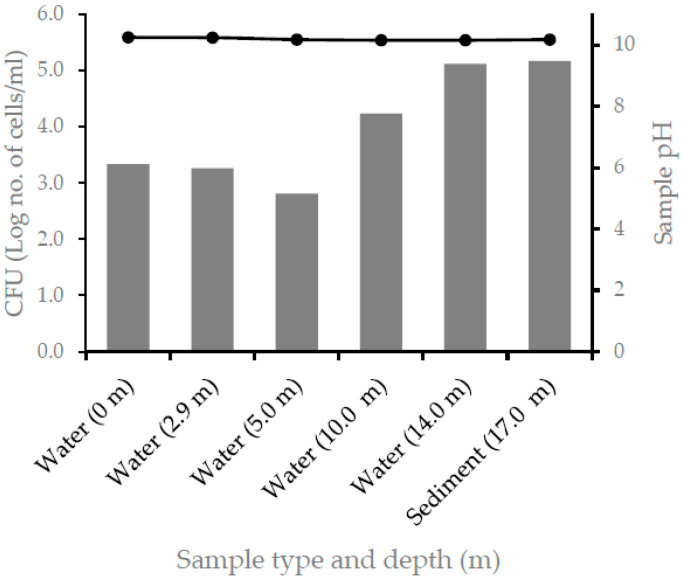
Colony-forming units (bar graph) from enrichment culture in a nitrogen-free media inoculated with samples collected from different depths. The pH of the samples taken from each depth is also shown with the line graph.

**Figure 2 microorganisms-10-01760-f002:**
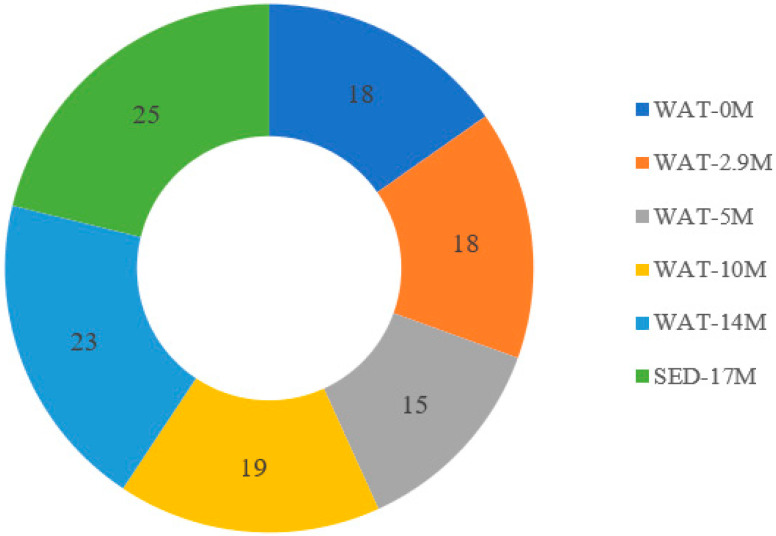
Number of nitrogen-fixing bacterial isolates obtained from each depth. WAT represents water sample and SED represents sediment sample.

**Figure 3 microorganisms-10-01760-f003:**
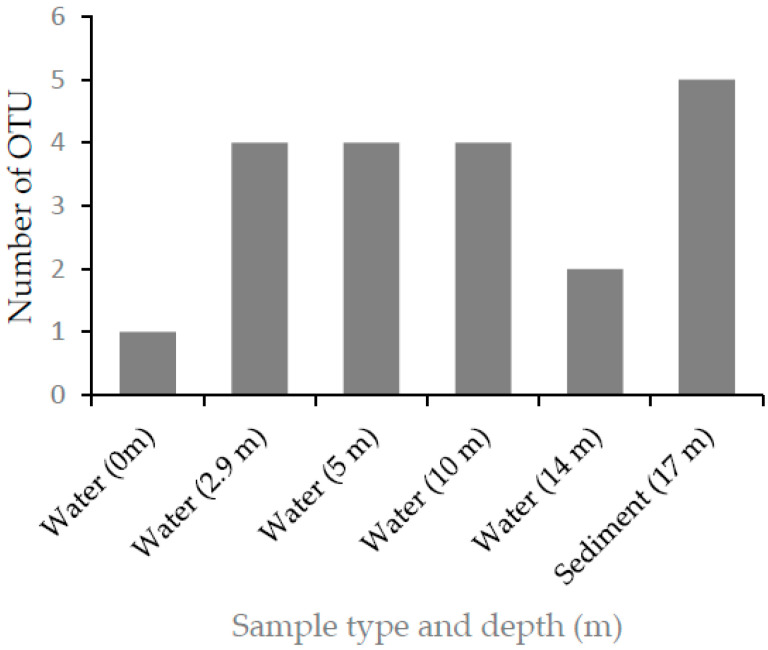
Depth distribution of nitrogen-fixing OTUs isolated from Lake Chitu.

**Figure 4 microorganisms-10-01760-f004:**
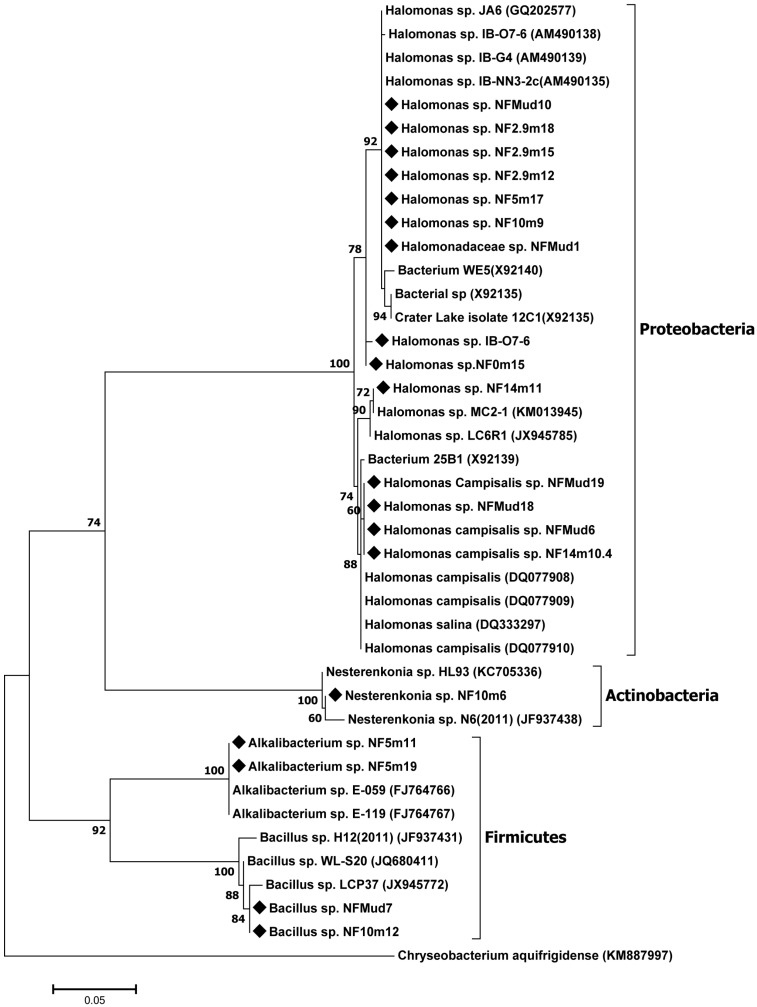
Phylogenic tree showing the evolutionary relation of nitrogen-fixing alkaliphilic bacterial strains. 16S rRNA gene sequence-based phylogenetic tree generated by using the maximum likelihood method showing the relationships between the strains studied and close matches. Numbers at nodes indicate percentages of occurrence in 1000 bootstrapped trees; only values greater than 50% are shown. Bar 0.05 number of substitutions per nucleotide position. *Chryseobacterium aquifrigidense (KM887997*) was used as out-group. Rhombuses indicate strains isolated in this study.

**Figure 5 microorganisms-10-01760-f005:**
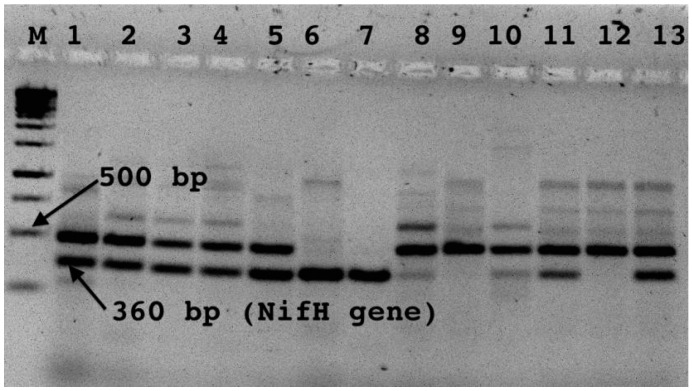
Result for presence *nifH* gene. **Key**: M: Marker, 1: NF0m15, 2: NF2.9m15, 3: NF2.9m18, 4: NF2.9m12, 5: NF10m9, 6: NF10m6, 7: NF14m10, 8: NF14m10.4, 9: NFM1.4, 10: NFM6.4, 11: NF7.4, 12: NFM18, 13: NFM19.

**Figure 6 microorganisms-10-01760-f006:**
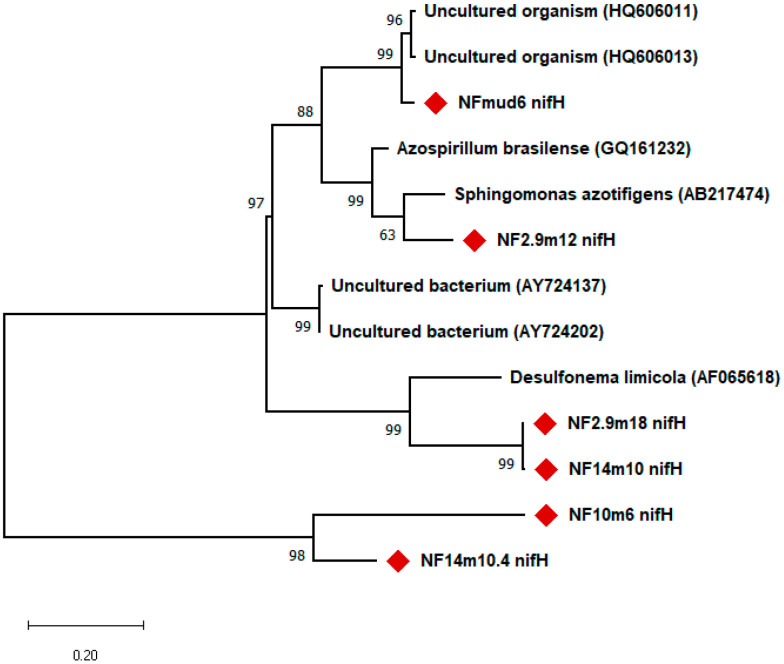
Phylogenetic tree showing representative *nifH* gene sequences among the diazotrophs. Bootstrap values greater than 50% (out of 100 replicates) are indicated at branch nodes. Bar 0.2 substitutions per nucleotide position.

**Table 1 microorganisms-10-01760-t001:** Sample types and physicochemical measurement data from Lake Chitu.

Sample Type	Depth (m)	pH	Oxygen Status	Salinity (%)	Conductivity mS/cm^−1^	Temperature (°C)
Water	0	10.25	Aerobic	5.5	64.3	24.8
Water	2.9	10.24	Transition	5.5	66.9	25.2
Water	5	10.18	Anoxic	5.5	62.5	24.2
Water	10	10.16	Anoxic	5.6	64.2	23.7
Water	14	10.16	Anoxic	5.6	63.1	24.9
Sediment	17	10.18	Anoxic	ND *	ND *	24.9

* ND = Not determined.

**Table 2 microorganisms-10-01760-t002:** Coculture of a nitrogen-fixing alkaliphilic isolate (NF10m6) and non-nitrogen-fixing strain, *Alkalibacterium* sp. 3.5*R1, in a nitrogen-free medium.

Culture Type	CFU/mL	
	Isolate NF10m6	*Alkalibacterium* sp. 3.5*R1
Monoculture	5.2 × 10^3^	0
Co-culture	4 × 10^2^	3 × 10^2^

## Data Availability

The data in this publication can be made available upon request to the first author.

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
