# Peer review of "Diversity of Culturable Alkaliphilic Nitrogen-Fixing Bacteria from a Soda Lake in the East African Rift Valley"

_microorganisms, 2022, doi:10.3390/microorganisms10091760_

Round 1
Author Response
Response to the Reviewers
Reviewer #1
Abstract:
Arthrospira fusiformis (= Spirulina platensis); Delete =
Authors response (AR): We accept the comment and deleted as suggested
High primary production requires supply of nitrogen, but the lake is a closed system with no external nutrient input?
AR: We accept the comment and in the revised manuscript the sentence is changed to read as follows. “High biomass accumulation requires adequate supply of nitrogen. However, Lake Chitu is a closed environment without any external nutrient input”.
In this study we tested if biological nitrogen fixation could supply nitrogen required in the lake. ?
AR: We accept the comment and in the revised manuscript the sentence is changed to read as follows “The aim of this study was to isolate culturable nitrogen fixing alkaliphiles and evaluate their potential contribution in the nitrogen economy of the soda lake”.
Line 19: nifH gene; nifH change to italic in whole manuscript
AR: We accept the comment and corrected as suggested
Line 20: sp.; Change to non-italic and 3.5*R1, what is it mean?
AR: We accept the comment and corrected as suggested
3.5*R1 is a strain code given by the researchers who isolated it first. It is a known non-nitrogen fixing strain which we used for co-culturing. We selected this strain because, on agar plates, it shows red coloration very distinct from all the nitrogen fixing isolates obtained in this study.
Line 21: both strains grew well; Which strains mentioned the name or code?
AR: We accept the comment the strain codes are given (please see below)
Line 21: But, when the non-nitrogen fixing strain was alone, there was no growth. Confusing this sentence means first how you know this is non-nitrogen fixing strains and where this was no growth.
AR: We accept the comment of the reviewer. In the revised manuscript the sentence was modified as follows. “When inoculated alone, Alkalibacterium sp. 3.5*R1 failed to grow on a nitrogen-free medium but grew very well when co-cultured with a nitrogen-fixing alkaliphile NF10m6 isolated in this study indicating availability of nitrogen”.
Line 22: This indicates release of reduced nitrogen by the nitrogen-fixing strain allowing the non-nitrogen fixer to grow. Again this sentence is confusing?
AR: We accept the comment and in the revised manuscript it is modified to improve clarity as indicated in our response above.
These results show that nitrogen fixation by alkaliphilic species may play an important role in the nitrogen economy of soda lakes.Economy of soda lakes means you are telling this lake is also an agriculture land and the purpose of this study to find an alkaliphilic strains to fix the nitrogen in crops. If this then revised the sentence otherwise clarify?
AR: We accept the comment and in the revised manuscript it is modified to improve clarity “These results show that nitrogen fixation by alkaliphiles may have an important contribution as a source of nitrogen in soda lakes”.
I find the abstract is not written well and does not reflect the content well and clear.
AR: We took the comment seriously and significantly improved the abstract, about half of it modified in the revised manuscript.
Introduction is not clear because the objective of this manuscript is:
The aim of this study was, therefore, to investigate the presence of culturable nitrogen fixing bacteria and study their diversity. But there is no sufficient literature to discuss the diversity of nitrogen fixing bacteria, author only focus on Lake Chitu, and length of introduction is also big.
In my opinion authors need to focus more on the diversity of microbes earlier reported and which types of microbes present highly.
AR: Based on this comment and we modified the aim of the study as follows. “The aim of this study was, therefore, to investigate the presence and diversity of culturable nitrogen fixing alkaliphiles and their potential role in in the nitrogen economy of the soda lake habitat”.
Methods:
Please indicate how many replicates per sample were collected, because if you are going for diversity analysis sample size is most important.
AR: Corrected as suggested
Statistical analysis is also missing in Table 1.
AR: The physical parameters shown in Table 1 were taken as indicators of the condition on the samples as indicators of the conditions in the samples to see if there was stratification. We thus did not see the need to do statistical analysis because we do not emphasize on the change in those physical parameters in the lake.
Another important drawback of your media enrichment, because if you are going for diversity analysis of any microbe’s minimum three to four different types of medium used.
Please justify.
AR: The aim of our study was to demonstrate that the presence or absence of nitrogen fixing alkaliphiles. To capture such organisms, we used a medium totally free of any nitrogen source. All required metal ions were included in the medium. We also used glucose as the carbon source because we believe it is carbon source that be used by most organisms. Therefore, as long as the medium we used a medium free of nitrogen we were not convinced varying the carbon source and the metal ions (the only components of the medium) could make much difference to isolate culturable potential nitrogen fixing alkaliphiles.
Plates were incubated at 30°C is also a questionable; please clarify.
AR: The temperature we measured at the lake was about 25°C. Therefore, on the other hand the temperature in the lab we worked on vary between around 10 during the night and as high as 25 during the day. Therefore, for better control of temperature of our incubator we used 30°C. We do not believe this is too far from the natural temperature these organisms are adapted.
Why authors are not used by Acetylene Reduction Assay (ARA) method to measure the nitrogenase activities because this method is mostly used and accurate quantify.
AR: We are aware of the Acetylene reduction assay method and we agree this would have been the most preferred method. However, we did not have access for the required equipment and thus preferred to use an indirect method. We believe the indirect method we used combined with the detection of the nifH gene could allow us toe evaluate nitrogen fixation.
PCR amplification and amplified ribosomal DNA restriction analysis (ARDRA)
This is also not a correct sequence for your study?
Firstly you go for 16S PCR amplification then why go for ARDRA
And if you go for ARDRA then why select only one enzyme because this method is used to differentiate the strains on the basis of band patter then go for dendrogram and after selected the representative strains.
AR: We used ARDRA to group the isolates and select representative isolates from each ARDRA group for sequencing because it is costly to sequence all the isolates. As indicated by the reviewer we used only one restriction enzyme, TaqI, for ARDRA grouping of the isolates for sequencing. However, we did not use the ARDRA grouping directly for construction of dendrogram, instead, representative isolates from each ARDRA group was taken for sequencing. Unless the ARDRA group (OTU) was unique, two or more representative isolates from the ARDRA groups were taken for sequencing. On top of that, our previous studies showed that TaqI is effective enzyme to differentiate alkaliphilic strains.
Sequencing of selected isolates; No need to write two times of primer sequence merge in above section
AR: We accept the comment and corrected as suggested
Reviewer 2 Report
The subject of the manuscript corresponds to the subject of the journal and it can potentially be accepted for publication after a minor revision of the text.
First of all, the Introduction should be improved. Please pay more attention to the problem of the nitrogen cycle in soda lakes instead of describing the minor features of these lakes. Please add to this section the hypothesis that you will test in this study.
L105-109: What was the water and sediment samples number (replications)?
Table 1: Is the unit of conductivity specified correctly?
Table 1: Is there any data on the content of mineral or total nitrogen in water and sediment samples?
L222: the "%" sign is probably missing
Fig. 1, 3: Is it possible to give not only the mean values, but at least the standard deviation or the means error? Or to conduct a more serious statistical analysis of the data? I think this is important for understanding the differences in the living conditions of bacteria at different depths.
L361-363: Obviously, it is necessary to edit this text
In general, when discussing the results, pay more attention to the potential mechanisms underlying the described phenomena.
L394: It is necessary to formulate a Conclusion. What new has this research yielded? What patterns did it reveal?
Author Response
The subject of the manuscript corresponds to the subject of the journal and it can potentially be accepted for publication after a minor revision of the text.
AR: We thank the reviewer for the positive comment.
First of all, the Introduction should be improved. Please pay more attention to the problem of the nitrogen cycle in soda lakes instead of describing the minor features of these lakes. Please add to this section the hypothesis that you will test in this study.
AR: As per the suggestion of the reviewer we made the necessary improvement in the introduction section.
L105-109: What was the water and sediment samples number (replications)?
AR: We acknowledge this detail was missing in our original submission but we now made a correction. It was triplicate samples.
Table 1: Is the unit of conductivity specified correctly?
AR: Here too we made the necessary correction
Table 1: Is there any data on the content of mineral or total nitrogen in water and sediment samples?
AR: In our study this was not measured because of time and resource limitation in our project. However, there are such data in the literature, and we hope those interested can easily access them.
L222: the "%" sign is probably missing
AR: We thank the reviewer for pointing out this and it is now corrected.
Fig. 1, 3: Is it possible to give not only the mean values, but at least the standard deviation or the means error? Or to conduct a more serious statistical analysis of the data? I think this is important for understanding the differences in the living conditions of bacteria at different depths.
AR: The aim of or study was not to see whether there are nitrogen fixers along at different depths and not to specifically evaluate differences among the different depths. As a result, we only show the mean values. In the future we plan to conduct more detailed study along the depth profile, including the uncultured organisms.
L361-363: Obviously, it is necessary to edit this text
In general, when discussing the results, pay more attention to the potential mechanisms underlying the described phenomena.
AR: We relooked at this section and made changes.
L394: It is necessary to formulate a Conclusion. What new has this research yielded? What patterns did it reveal?
AR: As we stated in the revised manuscript, we indicated the main findings of this study and its implications. We believe this will open more detailed studies on this and even allow practical applications. For example, in the last part we try to indicate the possibility of spirulina production using nitrogen fixation and CO2 released by industries. This is just to indicate to readers potential areas of research using such organisms.
Reviewer 3 Report
The aims of the study could be better described in the abstract.
Why did the authors only use one sample for the determinations in Table 1?
Section 2.4. should not be called "Genomic DNA extraction", since the authors do not really extract DNA; they use a cell lysate instead.
What is the size of the sequenced 16S rDNA fragment?
line 188: please correct to "...a fragment of the nifH gene of approximately 360 bp..." And why did the authors use such a small fragment of the gene?
Should gene names be in italics?
Why did the authors use Jukes-Cantor as the DNA substitution model for the reconstruction of both 16S rDNA and nifH phylogenetic trees? Why did they not run a Modeltest instead?
In Figure 4, the indication on the number of substitutions per nucleotide position is missing.
Is Figure 5 the original picture? Or have its colours been inverted? If so, please show the original gel photograph!
Figure 6: is this also a Maximum Likelihood tree?
Please include a conclusions section, and highlight the importance of the biodiversity in the lake and its contribution towards the Sustainable Development Goals (SDGs) of the 2030 Agenda of the United Nations.
The list of cited references is a bit too old. The authors could add more recent papers to their list of references.
Author Response
The aims of the study could be better described in the abstract.
AR: We accept the comment and in the revised manuscript we made changes to address these.
Why did the authors only use one sample for the determinations in Table 1?
AR: At the time of sampling, we wanted to check the physicochemical parameters and had these measurements. If multiple samples were taken there might have been some slight variation. However, our measurements directly correspond to previously measured values at different occasions and reported by different authors.
Section 2.4. should not be called "Genomic DNA extraction", since the authors do not really extract DNA; they use a cell lysate instead.
AR: We use modified CTAB method (Bengtsson et al.,2010) to extract DNA after we extract cell lysate. So the extract was not cell lysate, instead it was a genomic DNA pellete, isolated from pure bacterial isolates and suspended in TE buffer.
What is the size of the sequenced 16S rDNA fragment?
AR: The fragment size is 760bp
line 188: please correct to "...a fragment of the nifH gene of approximately 360 bp..."
AR: Corrected as suggested
And why did the authors use such a small fragment of the gene?
AR: We used specific nifH gene primers designed previously for specific amplification nifH gene of approximately 360 bp were used
Should gene names be in italics?
AR: Thank you very much for the comment. The gene names were corrected and written in italics as suggested
Why did the authors use Jukes-Cantor as the DNA substitution model for the reconstruction of both 16S rDNA and nifH phylogenetic trees? Why did they not run a Modeltest instead?
AR: The evolutionary history inferred by using the Maximum Likelihood method based on the Kimura 2-parameter model.
Round 2
Reviewer 1 Report
These are all interesting findings with valuable contribution and think this is meritorious of publication after carefully consideration of manuscript language and structure errors. Below you will find few examples that may be helpful in improving the manuscript.
Line 20: sp is non italic and change to sp.
Line 99: their potential role in in; please correct this and delete one (in).
Author Response
Line 20: sp is non italic and change to sp.
AR: Corrected a suggested
Line 99: their potential role in in; please correct this and delete one (in).
AR: Corrected as suggested
Reviewer 3 Report
The authors have satisfactorily replied to most of the comments made by the reviewers.
However, a few clarifications are still needed.
Why did the authors only use one sample for the determinations in Table 1?
AR: At the time of sampling, we wanted to check the physicochemical parameters and had these measurements. If multiple samples were taken there might have been some slight variation. However, our measurements directly correspond to previously measured values at different occasions and reported by different authors.
Precisely. There must be some variation that should be taken into account. If the authors do not have more values from their own samples then compare with the values from other others in Table 1.
What is the size of the sequenced 16S rDNA fragment?
AR: The fragment size is 760bp
Please include this information in the revised manuscript.
Why did the authors use Jukes-Cantor as the DNA substitution model for the reconstruction of both 16S rDNA and nifH phylogenetic trees? Why did they not run a Modeltest instead?
AR: The evolutionary history inferred by using the Maximum Likelihood method based on the Kimura 2-parameter model.
Please include this information in the revised manuscript. And explain why you chose this model. Did you run Modeltest?
Author Response
Precisely. There must be some variation that should be taken into account. If the authors do not have more values from their own samples then compare with the values from other others in Table 1.
AR: pH, salinity and conductivity measurements reported since 1994 is included under the results section (Line 229 to 233 in the revised document).
Please include this information in the revised manuscript.
AR: Yes, it is included in the revised manuscript
Please include this information in the revised manuscript. And explain why you chose this model. Did you run Modeltest?
AR: We Yes, the information is now included in the revised manuscript, highlighted in green under the materials and methods section, Line 213 – 214
Regarding running of the Modeltest, no we did not run model test to find the best fit evolutionary model. Instead, we used the Maximum likelihood method using distances calculated with the Kimura 2-parameter model, which, in our opinion is the most widely used model allowing corrections for transitional and transversional substitution rates, while assuming that the four nucleotide frequencies are the same and that rates of substitution do not vary among sites for construction of phylogenetic tree.